# Combining Ability and Reciprocal Effects for the Yield of Elite Blue Corn Lines from the Central Highlands of Mexico

**DOI:** 10.3390/plants12223861

**Published:** 2023-11-15

**Authors:** José Luis Arellano-Vázquez, Germán Fernando Gutiérrez-Hernández, Luis Fernando Ceja-Torres, Estela Flores-Gómez, Elpidio García-Ramírez, Francisco Roberto Quiroz-Figueroa, Patricia Vázquez-Lozano

**Affiliations:** 1Instituto Nacional de Investigaciones Forestales, Agrícolas y Pecuarias, Campo Experimental Valle de México, Coatlinchán 56250, Estado de México, Mexico; arellano.jose@inifap.gob.mx; 2Instituto Politécnico Nacional, Unidad Profesional Interdisciplinaria de Biotecnología, Av. Acueducto s/n, La Laguna Ticomán, Ciudad de México 07340, Mexico; esfloresg@ipn.mx (E.F.-G.); pvazquezl@ipn.mx (P.V.-L.); 3Instituto Politécnico Nacional, Centro Interdisciplinario de Investigación para el Desarrollo Integral Regional Unidad Michoacán, Justo Sierra 28, Jiquilpan 59510, Michoacán, Mexico; lfceja@ipn.mx; 4Universidad Nacional Autónoma de México, Facultad de Química, Av. Universidad y Copilco, Ciudad de México 04510, Mexico; elpidio@unam.mx; 5Instituto Politécnico Nacional, Centro Interdisciplinario de Investigación Para el Desarrollo Integral Regional, Unidad Sinaloa, Blvd. Juan de Dios Bátiz Paredes 250. Col. San Joachín, Guasave 81101, Sinaloa, Mexico; fquiroz@ipn.mx

**Keywords:** *Zea mays* L., diallel crosses, agronomic performance, blue grain, plant characteristics

## Abstract

The development of hybrid plants can increase the production and quality of blue corn, and, thus, satisfy its high demand. For this development, it is essential to understand the heterotic relationships of the germplasm. The objectives of this study were to determine the effects of general (GCA) and specific (SCA) combining ability, as well as the reciprocal effects (REs) on the yields of 10 blue corn lines, and to select the outstanding lines. Diallel crosses were generated with 10 lines and evaluated at the Valle de México Experimental Station in Chapingo, Mexico, and Calpulalpan, Tlaxcala, Mexico. There were differences (*p* ≤ 0.01) in the hybrids, Loc, effects of GCA, SCA, and REs, and in the following interactions: hybrids × Loc, GCA × Loc, SCA × Loc, and RE × Loc. For GCA, lines Ll, L4, L6, and L9 stood out, with significant values of 3.4, 2.9, 2.9, and 3.1, respectively. For SCA, the hybrids featured were L4 × L10, L2 × L10, L1 × L10, L7 × L8, and L2 × L6, with values of 3.0, 2.5, 2.3, 2.3, and 2.2, and yields of 11.2, 10.2, 10.4, 10.4, and 10.5 t ha^−l^, respectively. There were no significant REs in these lines. Considerable effects of GCA and SCA were detected; therefore, we concluded that native populations had favorable dominance and additive genetic effects that could be used to support the development of high-yielding lines and hybrids.

## 1. Introduction

Maize (*Zea mays* L.) is the world’s most important agricultural crop in terms of the area cultivated, the quantity of grain harvested, and the value of production [1]. It is the main food source in the diets of marginalized populations in Latin America, sub-Saharan Africa, Asia, and the Caribbean [2], providing the most calories (65–85%) and often representing the only source of protein [3,4].

Blue corn is important in Mexican cuisine, as it is the basic ingredient of many traditional dishes, due to its favorable nixtamalization characteristics, its culinary quality, and, of course, its attractive and appetizing color [5].

The blue coloring of the corn grain, with different intensities and shades, is caused by anthocyanins, plant pigments located mainly in the pericarp, the aleurone layer, or in both structures of the grain (2). Anthocyanins are polyphenols that constitute a large group of secondary metabolites with diverse organoleptic effects.

Recently, anthocyanins were reported to have nutraceutical, free radical scavenging, and anti-inflammatory properties, as well as aiding in the regulation of cell division; in addition, they help in the treatment of cardiovascular and neurodegenerative diseases [6,7], a property which has also led to a greater interest in consuming maize. This consumer preference has caused the actual production, based on creole genotypes, to not satisfy the growing demand for this type of pigmented maize. It is estimated that the consumption demand of this grain in traditional foods exceeds the 450 thousand tons per year that are currently produced. It is, therefore, appropriate to develop hybrids and improved varieties of blue maize, with higher yields and better plant and grain characteristics. To identify the best lines and crosses for generating hybrids with the desired characteristics, it is important to determine the effects of the combining abilities in blue corn lines from the highlands of Mexico. Sharing the knowledge gained from this study with other countries could lead to an expansion in the use of the genetic diversity of blue corn to develop new hybrids. These new hybrids could open up the possibility of producing foods with new physical and nutraceutical properties.

In the Central Highlands of Mexico, the genetic diversity of blue corn is extensive; however, there are few studies related to genetic improvements in this variant of corn to increase its yield and resistance to lodging, or to optimize its grain texture. Native creole varieties of blue corn yield from 2.7 to 6.6 t ha^−1^ [8], and it is feasible to exceed this yield by developing hybrids that are suitable for rainfed and irrigated areas. The formation of hybrids that will be commercially released after being evaluated in multiple environments involves identifying populations or varieties with broad genetic bases, developing lines, defining their general and specific combining abilities, and selecting superior lines [9].

The general combining ability (GCA) and specific combining ability (SCA) concepts are essential for developing hybrids. The first defines the average behavior of a line in its hybrid combinations, and the second evaluates hybrid combinations with respect to the average behavior of the lines. GCA results from additive gene action, while SCA depends on dominance and epistasis [10]. When the effects of SCA predominate, it is recommended to exploit them through hybridization [11]. Information on combining ability and heterosis is imperative in a genetic improvement program to develop hybrids or synthetic varieties [12].

Studies on the effects of combining ability and heterosis on yield in tropical germplasm lines indicated that GCA was more important than SCA [13,14,15,16,17], while other authors [18] found that only the effects of SCA were significant, while it has also been determined that both GCA and SCA were significant for grain yield [19].

It is known that the variances and reciprocal effects of SCA predominate over those of GCA [20], which suggests that, with the determination of reciprocal effects, it is possible to define the order of the female parental lines according to their productive capacity [21].

In corn populations from the High Valleys of Mexico, high GCA values were associated with high yield [22]. Lines from the same region, derived from the Mich 21 and Tlax 151 populations, showed GCA values of 0.92, and their SCA cross values ranged from 1.1 to 1.7 [23]. Furthermore, lines from these same populations showed SCA values of 1.98 [24], and combinations between lines of the Mich 21 population, with different levels of inbreeding, showed GCA values of 0.79 and SCA values of 2.4 [25].

Meanwhile, temperate–subtropical maize lines, evaluated in high valleys, had GCA values of 1.5 and SCA values of 2.0 [26]. Maize lines with different pericarp colors, derived from native populations, had a GCA value of 1.2 and an SCA value of 0.85 for yield [27].

There are few studies on inbred lines of blue corn from the Central Highlands of Mexico that show the genetic effects of combining abilities and reciprocal effects, as well as their application in a commercial hybrid development program. Therefore, the objectives of this study were as follows: (1) to determine the effects of general and specific combining abilities, as well as the reciprocal effects for the yields of 10 elite lines of blue corn from the Central Highlands of Mexico; and (2) to select lines with outstanding genetic effects to form hybrids and synthetic varieties of blue corn.

## 2. Materials and Methods

### 2.1. Genetic Materials

Ten elite inbred lines of blue corn, generated by the Genetic Improvement Program of Blue Corn of the High Valleys of the National Institute of Forestry, Agricultural, and Livestock Research (Instituto Nacional de Investigaciones Forestales, Agrícolas y Pecuarias, INIFAP, Mexico) were used. These lines were derived from the F_2_ crosses between the blue corn variety “Cocotitlán 22”, of the Chalqueño variety [28,29], and a hybrid of white grain, from the same region, as a source of resistance to lodging, and from the cross between the blue corn varieties “Nexapa” (Chalqueño variety) and Oaxaca 6867 (Bolita variety) [28,30]. The lines had five generations of self-fertilization and per se selection. The Chalqueño varieties were used as sources of yield and intense blue color in the aleurone layer of the grain, and the Bolita variety was chosen for its low plant stature, resistance to lodging, and grain characteristics, including its semicrystalline texture and blue-purple coloring (Table 1).

To derive the lines used in this study, the introgression of exotic germplasm from the “Bolita” variety, which has a dry semitropical geographical origin, into the germplasm of the “Chalqueño” variety, with a temperate–cold origin, was implemented. Genetic characteristics are caused by the geographic origins of populations and, thereby, increase the possibility of heterotic responses [31]. Direct and reciprocal diallel crosses between the lines were performed using Griffing’s method 1 model 1 [32]. Ten progenitor lines were added to ninety crosses, and a statistical evaluation was performed to determine the indicated genetic effects.

Lines L1 to L5 were derived from the cross between plants of the creole variety “Cocotitlán”, which served as the source of blue grains, and hybrid white grain resistance to lodging (both belonging to the “Chalqueño” variety). Meanwhile, lines L6 to L10 were derived from the cross between the creole varieties “Nexapa” and “Oaxaca” with blue grains, belonging to the “Chalqueño” and “Bolita” varieties, respectively (Table 1).

### 2.2. Evaluation Locations, Sowing Dates, and Agronomic Management

The experiments were established in 2016 in two localities (Loc): (a) the Valle de México Experimental Station (Chapingo, Mexico), located at 19°29′ LN, 98°53′ LW, and 2240 m above sea level, with a temperate climate, precipitation in summer, and annual average precipitation and temperature of 643 mm and 15.1 °C, respectively; (b) Calpulalpan, Tlaxcala, located at 19°35′ LN, 98°34′ LW, and 2583 masl, with annual average precipitation and temperature of 645.4 mm and 14.2 °C, respectively [33].

The process of preparing the soil involved subsoiling to plow the ground to a depth of 0.5 m, in order to break up the clay profile. This was followed by two plows to a depth of 0.3 m to loosen and furrow the soil, with 0.8 m between each furrow. Sowing involved depositing two seeds every 16 cm, and 20 d after seedling emergence, one plant was removed to ensure that there were only 32 plants within the 5 m of the furrow. To control broadleaf weeds, the commercial herbicide “Primagram Gold” (atrazine 374 g L^−1^ + s-metolachlor 290 g L^−1^, active ingredients) was applied one day after irrigation for seed germination at a rate of 2 L ha^−1^ dissolved in 200 L of water. After 40 d, in the intermediate vegetative development stage, the second application of herbicide was conducted, with a mixture of the products “Marvel” (dicamba 132 g L^−1^ + atrazine 252 g L^−1^, active ingredients) and “Hierbamina” (2,4-D, active ingredient, 479 g/L), at a rate of 1 L ha^−1^ of each, dissolved in 200 L of water. *Spodoptera frugiperda*, *Frankliniella* spp., and *Geraeus senilis* insects, were controlled using Karate Zeon insecticide (lambda cyalotrine, active ingredient, 50 g/L^−1^) was applied (300 mL/ha^−1^ dissolved in 300 L of water). Irrigation was first applied to promote germination (on May 8 in Chapingo and on April 29 in Calpulalpan), then again 75 d after sowing in Chapingo and 95 d in Calpulalpan, and once again during the grain-filling stage (115 d after planting in Chapingo and 135 in Calpulalpan). The population density of the experimental specimens was 65 thousand plants per hectare. The fertilization doses were 120, 60, and 30 kg ha^−1^ of nitrogen, phosphorus, and potassium, respectively. Harvesting took place between November 15 and December 15.

### 2.3. Recorded Data

The parameter days to tassel (Tassel) was recorded as the period (d) elapsed, from the day of the germination irrigation application to the day on which 50% of the plants in the experimental plot presented pollen dispersion from the main branches of the recorded inflorescence. The plant height (PH) was measured as the distance (cm) from the base of the plant to the base of the male inflorescence. The grain yield (Y, ton ha^−1^) was calculated from the weight of the ears harvested per experimental plot, adjusted to an 80% grain content in the ear and a 14% grain moisture content. The test weight (TW) was obtained using the methods described by the authors of [34]. A scale of 5 to 10 was used for the visual grading of the grain color (COLOR), where 5–6 corresponded to purple, 7–8 to dark blue, and 9–10 to black.

### 2.4. Experimental and Genetic Designs and Statistical Analyses

A 10 × 10 lattice experimental design, with two repetitions, was used in the field experiments, which included 45 direct crosses, 45 reciprocal crosses, and 10 parental lines of these crosses as treatments. The 100 treatments were randomly arranged in 10 sub-blocks of 10 treatments each for each repetition, established in field experiments at two locations. The diallel crosses were statistically analyzed using Griffing’s method I model 1 [32] to determine the effects of general and specific combining abilities and reciprocal effects. For the statistical analysis of the experimental data, Zhang and Kang’s proposed protocol [35] (pp. 1–19) and the statistical analysis computer software SAS [36] were used.

## 3. Results

### 3.1. Analysis of Variance of Diallel Crosses

In the combined analysis of variance for yield (Table 2), differences (*p* ≤ 0.01) were detected in all sources of variation, except for the RE × Loc interaction. The variability in yield was concentrated in hybrids and SCAs, since they represented 68 and 62% of the total sum of squares, respectively. Furthermore, the variability in SCAs was 39% higher than that in GCAs. The significance (*p* ≤ 0.01) detected for GCA × Loc, SCA × Loc, and RE × Loc indicated that these effects were not consistent across localities, i.e., each presented different environmental conditions.

### 3.2. General Combining Ability

In the estimation of the genetic parameters, the effects of GCA (Table 3) were significant (*p* ≤ 0.01) for all the lines except L10. GCA values of 1.61 to 3.46 were observed, which is important because the lines were generated from native populations of the high valleys without intrapopulation genetic improvement. Lines L1, L4, L6, and L9, with GCA effects of 3.46, 2.98, 2.90, and 3.13, respectively, exhibited the best expression of additive effects; however, their yields were modest. Due to their agronomic behavior and test weights, the lines were of the late type, medium size, and floury–crystalline grain.

### 3.3. Specific Combining Ability and Reciprocal Effects

There was statistical significance for SCA in 66% of the direct crosses and 28% of the reciprocal crosses (Table 4). The direct crosses between the lines L1, L2, L4, L6, L8, and L9 × L10 were significantly associated with high SCA, with values of 2.3, 2.5, 3.0, 1.8, 1.7, and 1.5, respectively. The direct crosses of lines L1, L2, L3, L4, and L5 × L9 were also significant, with SCA values from 0.8 to 1.9. It was notable that the reciprocal crosses of L10 (as a female parent) with the rest of the lines resulted in significant but negative SCA values; therefore, there were no reciprocal or maternal effects on performance in these reciprocal crosses. The few statistically significant reciprocal effects of SCAs, in lines L7 and L8, were considered a marginal contribution to the results of this study (Table 4).

According to Vasal et al.’s [37] proposal for designing heterotic groups, and based on the significant effects of SCA, the heterotic pattern established in this research was represented in lines L1, L2, L4, and L5, which were derived mainly from the “Chalqueño” variety and their combination with line L10, which was formed from the varieties “Chalqueño” and “Bolita”.

### 3.4. Yield and Agronomic Characteristics of Hybrids

In the statistical analysis of the complete set of hybrids, differences (*p* ≤ 0.01) were found for the characteristics Y, Tassel, and PH among Loc, hybrids, and hybrids × Loc. For COLOR, significant results were found only among hybrids (Table 5), which evidenced that the environmental conditions of the localities were different, that the hybrids showed genetic variability in yield and plant characteristics, and that the genotypes responded differently in their characteristics according to environmental expression.

The hybrids with significantly higher yields had values from 10.2 to 11.4 t ha^−1^ (Table 6, summarized for the hybrids with the highest yields). The results highlight that lines L1, L2, L4, L9, and L10 were progenitors of the hybrids with yields that exceeded 11.0 t ha^−1^. Furthermore, it was observed that the performances of the direct crosses, L1 × L5, L1 × L6, L2 × L6, L2 × L9, and L2 × L10, were similar to those of their reciprocals, i.e., there were no reciprocal and/or maternal effects.

Based on the characteristics Tassel, PH, TW, and COLOR, the hybrids showed late male flowering and a tall plant size, most with a semicrystalline grain texture (with a TW from 69 to 73 kg hL^−1^) and dark blue coloring.

## 4. Discussion

The variability in grain yield was concentrated in hybrids and SCAs, since they accounted for 68% and 62% of the total sum of squares, respectively. This finding is consistent with the results reported by the authors of [17]. Additionally, the variability in SCAs was found to be 39% higher than that of GCAs. The significance of GCA × Loc, SCA × Loc, and RE × Loc indicated that these effects varied between locations, which was also noted by the authors of [17,25].

The performance of the corn hybrids was mainly determined by the genetic effects of dominance or interaction, agreeing with the findings reported by several authors [16,18]. The high and significant values of GCAs in the lines studied corroborated that lines developed from populations with a broad genetic base showed favorable additive effects in their combinations, and the significant values of SCA indicated the possibility of establishing a favorable heterotic pattern through which to develop commercial hybrids [38]. The significant results detected for GCA × Loc, SCA × Loc, and RE × Loc indicated that the values for these effects were not consistent across locations. Due to the differences in environmental conditions that prevailed between localities, an intense genotype × environment interaction was observed, as documented in the literature [19]. Of the reciprocal crosses, 93% were not statistically significant. Therefore, it can be interpreted that no maternal effects manifested in most of the lines studied. However, the significance of REs and MEs could be attributed to some female progenitor lines, whose performance could have a cytoplasmic influence [21]. Although the reciprocal crosses had higher yields than the direct crosses, this does not necessarily indicate that the differences in the lines’ behaviors are due to the cytoplasmic effects observed in their hybrids [39]; therefore, the determinations of reciprocal and maternal effects on yield were not conclusive, coinciding with the common axiom in the analysis of diallel crosses of corn that established the absence of maternal effects for this trait [40].

Per the proposal to designate heterotic groups [37] and based on the significance of the effects of SCAs, it can be noted that the main heterotic pattern of this research was represented in the lines (L1, L2, L4, and L5) derived from the “Chalqueño” variety population, and its combination with the L10 line derived from the combination of the “Chalqueño” and “Bolita” populations. The effects of SCAs in these combinations were positive and significant, which implied that the lines were in opposite heterotic groups, as it has been documented that, when the effects of SCA are negative, the lines belong to the same heterotic group [37,41].

## 5. Conclusions

Considerable effects of general and specific combining abilities were observed for lines and crosses, which infers the existence of additive and dominance genetic effects in lines that allowed the generation of high-yielding experimental hybrids from the High Valleys of Mexico.

The combinations of lines generated from the germplasm sources of the populations of the Chalqueño and Bolita varieties showed the possibility of a favorable heterotic pattern for generating blue corn hybrids with good yield and plant height, as well as excellent intense blue coloration, present in the aleurone layer of the grain.

Combining ability studies in corn have mainly been carried out on white and yellow corn. However, the present study was conducted with blue corn due to the significant genetic diversity of this type of corn in Mexico. There is a lack of knowledge about the genetic effects of combining abilities in blue corn, and the present study aimed to shed light on them. The findings of this study can open up the possibility of expanding the use of the genetic diversity of blue corn in other countries in order to generate new hybrids and cultivars. This, in turn, will allow for the diversification of foods, with better nutraceutical properties due to their anthocyanin contents.

## Figures and Tables

**Table 1 plants-12-03861-t001:** Genealogies, parents, silking, and grain characteristics of inbred lines of blue corn from the Central High Valleys of Mexico.

Line	Genealogy	Parents	Silking (d)	Grain	
				Color	Texture
L1	BXCC-8-7-1-2-1	Cocotitlán 22 × Hybrid	85	Purple	Semicrystalline
L2	BXCC-3-1-3-6-4	Cocotitlán 22 × Hybrid	85	Blue	Semicrystalline
L3	BXCC-3-1-3-2-1	Cocotitlán 22 × hybrid	85	Purple	Semicrystalline
L4	BXCC-3-8-3-4-2	Cocotitlán 22 × Hybrid	85	Blue	Floury
L5	BXCC-2-1-5-2-6	Cocotitlán 22 × Hybrid	85	Blue	Floury
L6	NXOAX-168-2-1-2-2	Nexapa × Oaxaca	92	Blue	Floury
L7	NXOAX-51-1-1-2-1	Nexapa × Oaxaca	90	Blue	Floury
L8	NXOAX-46-l-1-1-1	Nexapa × Oaxaca	90	Blue	Semicrystalline
L9	NXOAX-28-2-3-4-2	Nexapa × Oaxaca	88	Blue	Semicrystalline
L10	NXOAX-19-5-l-1-2	Nexapa × Oaxaca	88	Blue	Semicrystalline

**Table 2 plants-12-03861-t002:** Analysis of variance for yields of diallel crosses of blue corn in two localities in the Central High Valleys of Mexico.

Variation Source	df	Sum of Squares	Mean Squares	F-Value	
Loc	1	654.3	654.3	181.7	**
Repetitions (Loc)	2	23.4	11.7	3.3	*
Hybrids	99	2788.7	28. I	7.2	**
Hybrids × Loc	99	361.3	3.6	2.5	*
GCA	9	1830.7	203.4	138.0	**
SCA	45	2551.1	56.6	38.4	**
GCA × Loc	9	57.0	6.3	4.3	**
SCA × Loc	45	228.7	5.1	3.5	**
RE	45	1965.1	43.6	29.6	**
RE × Loc	45	144.0	3.2	2.1	ns
ME	9	1769.4	196.6	133.4	**
Error	198	291.7	1.47		
Total	399	4119.7			

df—degrees of freedom, Loc—locality, GCA—general combining ability, SCA—specific combining ability, RE—reciprocal effect, and ME—maternal effect; *—significant (*p* ≤ 0.05), **—highly significant (*p* ≤ 0.01), and ns—not significant.

**Table 3 plants-12-03861-t003:** Effects of general combining ability (GCA), yield, and plant and grain characteristics of blue corn lines from the Central High Valleys of Mexico.

Line	GCA	Y (t ha^−1^)	Tassel (d)	PH (cm)	TW (kg hL^−1^)
L1	3.46 **	1.6 a	93 b	155 c	70 a
L2	2.80 **	1.2 a	92 c	170 b	70 a
L3	1.61 **	1.4 a	93 b	160 c	69 a
L4	2.98 **	2.4 a	93 b	180 a	71 a
L5	2.45 **	2.5 a	95 b	150 c	69 a
L6	2.90 **	1.5 a	99 a	180 a	70 a
L7	2.47 **	1.2 a	94 b	140 d	68 a
L8	2.38 **	1.7 a	96 a	180 a	68 a
L9	3.13 **	2.0 a	93 b	180 a	68 a
L10	−2.00 **	2.4 a	95 b	180 a	69 a
Standard error	0.857				
LSD		1.95	3.0	6.0	12

Y—grain yield, Tassel—male flowering, PH—plant height, and TW—test weight; **—highly significant (*p* ≤ 0.01) and LSD—least significant difference (*p* ≤ 0.05). letters in columns—significant differences (*p* ≤ 0.05).

**Table 4 plants-12-03861-t004:** Effects of specific combining ability (SCA) and reciprocal effects on yield of blue corn lines from the Central High Valleys of Mexico.

Lines										
♀\♂	L1	L2	L3	L4	L5	L6	L7	L8	L9	L10
L1		1.2 **	1.0 **	−1.0 **	2.0 **	1.3 **	0.8 *	0.7	0.8 *	2.3 **
L2	−0.6		−2.6 *	1.4 **	−0.5	2.2 **	0.9 *	1.1 **	1.9 **	2.5 **
L3	−0.8	0.04		0.5	−0.3	−0.5	1.9 **	−0.04	0.8 *	−1.3 *
L4	0.2	−0.6	−0.5		1.2 **	0.9 **	−0.1	−0.02	1.4 **	3.0 **
L5	0.3	0.2	0.2	0.0		1.2 **	1.4	0.0	0.9 *	1.1 **
L6	−0.5	−0.6	0.2	0.5	−0.2		1.0 **	0.4	0.3	1.8 **
L7	−1.2	0.08	−0.3	0.5	1.2 **	0.4		2.3 **	0.4	−0.3
L8	0.7	1.2 **	−0.4	−0.8 *	0.1	1.3 **	0.1		0.4	1.7 **
L9	−0.5	0.04	0.0	−0.1	−0.6	−0.8	0.4	−0.6		1.5 **
L10	−26 *	−26 *	−31 **	−26 **	−27 **	−26 *	−26 *	−27 *	−25 *	

*—significant (*p* ≤ 0.05), **—highly significant (*p* ≤ 0.01); direct effects of SCA above the diagonal line, with standard error = 1.03; reciprocal effects below the diagonal line, with standard error = 1.21.

**Table 5 plants-12-03861-t005:** Mean squares of the analysis of variance for yield, as well as plant and grain characteristics of the 100 blue corn hybrids (complete set) in two localities of the Central High Valleys of Mexico.

Variation Source	df	Y	Tassel	PH	TW	Color
Loc	1	859,758,220 **	20,302.0 **	0.7 **	2646.2 ns	2.5 ns
Blocks	19	1,999,839	8.4	0.2	1292.1	1.0
Hybrids	99	21,244,072 **	9.7 **	0.2 **	1165.5 ns	2.9 **
Hybrids × Loc	99	3,461,724 **	4.6 **	0.1 **	1132.4 ns	0.9 ns
Error	180	994,307	2.4	0.03	1088.7	0.9
C.V. (%)		11.6	1.7	8.0	45.4	12.0

df—degrees of freedom, Y—grain yield, Tassel—male flowering, PH—plant height, TW—test weight, color—grain color, Loc—localities, and C.V.—coefficient of variation; **—highly significant (*p* ≤ 0.01) and ns—not significant.

**Table 6 plants-12-03861-t006:** Yield, plant, and grain characteristics of the blue corn hybrids with the best average performances in two localities of the Central High Valleys of Mexico.

Hybrid (Cross)	Y (t ha^−1^)	Tassel (d)	PH (cm)	TW (kg hL^−1^)	Color
2 (L2 × L1)	11.4	89	250	72	8.5
10 (L6 × L1)	11.4	89	260	67	8.3
7 (L1 × L5)	11.3	86	220	69	7.2
26 (L6 × L2)	11.3	87	250	70	9.9
8 (L5 × L1)	11.2	88	230	69	7.7
59 (L4 × L10)	11.2	88	250	67	7.2
31 (L2 × L9)	11.1	87	220	72	8.5
32 (L9 × L2)	11.0	87	230	73	8.8
12 (L7 × L1)	10.9	87	250	74	8.8
16 (L9 × L1)	10.9	89	240	73	8.0
22 (L4 × L2)	10.9	87	240	71	9.9
29 (L2 × L8)	10.7	87	230	70	9.1
57 (L4 × L9)	10.6	86	230	70	7.4
58 (L9 × L4)	10.6	86	250	69	8.0
25 (L2 × L6)	10.5	87	230	73	9.4
60 (L10 × L4)	10.5	88	250	62	8.0
13 (L1 × L8)	10.4	87	250	71	8.3
79 (L7 × L8)	10.4	90	230	72	9.9
4 (L3 × L1)	10.3	86	230	71	8.5
9 (L1 × L6)	10.3	87	250	68	8.0
18 (L10 × L1)	10.3	88	250	67	7.7
88 (L10 × L8)	10.3	89	250	68	8.8
34 (L10 × L2)	10.3	88	240	71	9.6
51 (L4 × L6)	10.3	89	230	67	9.9
33 (L2 × L10)	10.2	89	240	71	6.6
80 (L8 × L7)	10.2	90	230	74	9.9
LSD	1.95	3.0	6.0	12	1.8

Y—grain yield, Tassel—male flowering, PH—plant height, TW—test weight, color—grain color, and LSD—least significant difference (*p* ≤ 0.05).

## Data Availability

The authors declare that all data supporting the findings of this study are available within this article.

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
