# Peer review of "Combining Ability and Reciprocal Effects for the Yield of Elite Blue Corn Lines from the Central Highlands of Mexico"

_plants, 2023, doi:10.3390/plants12223861_

Round 1

Reviewer 1 Report

Comments and Suggestions for Authors

The present manuscript “Combining ability and reciprocal effects for yield of elite blue corn lines from the Central Highlands of Mexico” is written well. However, there is no novelty in this manuscript it is simply giving the insights at regional scale which is only a main novelty of this manuscript.

Line 23: at two localities (Loc), here add information about the localities.

In the abstract section, also add information about the lines which ten lines were used.

Lines 28-30: (There were no significant RE in the lines. Considerable effects of GCA and SCA were detected. Native populations had favorable dominance and additive genetic effects that could be used to support the development of high-yielding lines and hybrids). These lines give no clear information, here authors must add some appropriate information and also a conclusion at the end of the abstract section.

I am not satisfied with the first 3 lines of the introduction section. The quality of the English language is very poor here, therefore, please revise these lines.

The introduction must be written at the start of the art form rather than adding simple information.

I also suggest the authors to improve the readability of the text starting from the abstract throughout the conclusion.

Add the parents of all lines in Table 1.

What about soil preparation and plant protection measures, this information can be added in MM sections.

I am satisfied with the discussion section, it should be written well with logical reasoning.

Overall, I am not satisfied with the present manuscript and there is no novelty in this study therefore, I will go with a negative decision.

Comments on the Quality of English Language

Moderate changes are needed. 

Reviewer 2 Report

Comments and Suggestions for Authors

1. This study showed the possibility of favorable heterotic pattern for the generation of blue  corn hybrid  with good yield, high plant quality and excellent intense blue coloration present the aleurone layer of the grain, so it can be applied in farm.

2. The results don’t to determinate of reciprocal and maternal effects on yield.

3. The references should be with  more recently 3 years.

Reviewer 3 Report

Comments and Suggestions for Authors

This study describes the diallel (not diallelic) analysis of blue corn inbreds in two environments. Some comments:

GCA and SCA effects are effects and do not have units. So you cannot say that GCA and SCA values are ton per ha. Correct this throughout. Please see a paper by Fasahat et al. (2016): Principles and utilization of combining ability in plant breeding. Biom. Biostat. Int. J. 4(1): 1-22, which very clearly explains the principles of GCA and SCA. Likewise SCA has nothing to do with genotype by environment interaction but is rather due to an interaction effect between two specific genotypes, or inbreds in this case. 

Then can you just clarify the statistical layout, you say you used a double lattice but it is not clear exactly what you mean by this. How many replications were there? In Table 2 it seems as though there are 3 reps, but in table 5 you show 20 blocks but no reps? Can you just clarify this? Only two environments give a very limited dataset. The results would have been much more conclusive with more environments and/or seasons. 

The annotated pdf is attached. 

Comments on the Quality of English Language

There are some minor English editorial corrections that need to be done. These are indicated on the pdf.

Reviewer 4 Report

Comments and Suggestions for Authors

Comments on the Quality of English Language

Round 2

Reviewer 1 Report

Comments and Suggestions for Authors

NA

Comments on the Quality of English Language

Minor changes are needed. 
